# Relationship between customer knowledge management and the value co-creation of fitness application customers: Mediating role of flow experience

**Chao Wang**[1], **Zhigang Wang**[2,3], **Liandi Liu**[4], **Kai Hua**[1,3]*

**1** Football College, Wuhan Sports University, Wuhan, Hubei, China, **2** Economics and Management College, Wuhan Sports University, Wuhan, Hubei, China, **3** Hubei Sports Industry Research Center, Wuhan, Hubei, China, **4** Renmin Hospital, Hubei University of Medicine, Shiyan, China

* 95216294@qq.com

**Data Availability Statement:** All relevant data are within the paper and its supporting information files.

**Funding:** This work is funded by the outstanding young and middle-aged scientific and technological

## Abstract

### Purpose

This study examined the impacts of customer knowledge management and flow experience on customer value co-creation and the mediating role of flow experience in the context of fitness apps.

### Design/methodology/approach

Using the questionnaire star platform to edit the questionnaire and collect data(n = 450). A structural equation modeling test was conducted to examine the relationships between the variables.

### Findings

The findings reveal that in a fitness app service scenario, customer knowledge management has a significant positive impact on customer flow experience, customer flow experience has a significant positive impact on customer value co-creation, and customer flow experience plays a partial mediating role in the path from customer knowledge management to customer value co-creation.

### Practical implications

The results could help fitness-app-related enterprises or service organizations understand the factors influencing and processes of customer participation in value co-creation and thus could help such enterprises and organizations formulate effective marketing strategies to realize customer value co-creation and ultimately to achieve their development goals.

### Originality/value

Using value co-creation theory and customer-dominant logic, this study analyzed the effects of customer knowledge management, flow experience, and customer value co-creation in

innovation team of Hubei Provincial Department of Education (T2023031), the major project of philosophy and social science research in colleges and universities in Hubei Province (22ZD120), and the key project of philosophy and social science of Hubei Provincial Department of Education (23D093).

**Competing interests:** The authors have declared that no competing interests exist.

the context of fitness apps and examined the mediating role of flow experience. The findings fill a gap in the theoretical research regarding customer value co-creation in the context of fitness apps and expand the scope of research on customer knowledge management and flow experience.

## 1. Introduction

With increasing affluence and given the impact of the COVID-19 pandemic, Chinese people have become increasingly health conscious; thus, fitness apps have proliferated to meet these health needs [1]. According to QuestMobile, the number of downloads of fitness apps, such as Keep, Nike+, Daily Yoga, and Pleasure Circle, has increased sharply. As of February 2021, 780 million users utilized "cloud sports" technology on fitness apps, and nearly 60 million people used vertical fitness apps [2]. Thus, China's mobile online fitness industry has high market potential, and the development of fitness apps that fulfill specific fitness needs of the Chinese population in the market is a major priority for app enterprises.

Studies on the development of fitness apps have investigated the factors influencing the willingness of users to use these apps and have adopted the models of technology acceptance [3], technology preparation [4], and technology integration to do so [5]. However, the success of a fitness app is not only dependent on user adoption. Fitness apps promote customers to be more active, connected, and informed; these apps can help companies gain a comprehensive understanding of their target market and can connect like-minded users [6]. Thus, the crucial role of customer value co-creation in the development of fitness apps should be should investigated, especially with regard to the development of new technologies or fitness methods.

The mechanisms underlying the effects of customer value co-creation on business development have been investigated, especially in the context of virtual brand communities and online platforms [7]. Studies have investigated the influence of customer interaction [8], innovation, customer engagement [9], user experience [10], and media sociality [11] on customer value co-creation. In the fitness app market, social media interactions play a crucial role in value co-creation, which is reflected in customer experience [12]. Customers not only wish to share their opinions with other customers but also want the company to consider and respond to their feedback. Such feedback results from customer knowledge [13]; thus, customer knowledge management is necessary to improve customer experience and to facilitate subsequent value co-creation, which enhances the app's overall value [14].

Under customer-led logic, value co-creation is a dynamic and social process that is based on company–stakeholder interactions, and it can promote mutual trust and communication between the company and its stakeholders [15]. In particular, the experience, knowledge, and ideas shared by customers on online platforms can be leveraged by enterprises to more effectively serve meet customer needs [16]. Thus, customer experience is central to the perceptions and subsequent actions of customers, especially in value co-creation.

In fitness, flow experience is key to customer experience, and flow can be affected by the form of participation and the type of media used [17]. Customers have a better perception of and participate more actively in fitness apps that help them attain the flow state more easily during exercise, thus aiding in value co-creation [18]. Various exercises, such as dancing, cycling, and yoga, can be integrated on apps to help customers achieve flow and have a more favorable fitness experience overall. Attaining such flow leads customers to be more

relationally and emotionally invested in the app and its community; thus, they are more likely to share their fitness journey and achievements with other users online.

As a product that integrates the social media interaction and sports fitness, fitness apps have become an important platform for customer value creation [11]. Through diverse interaction channels, such as virtual brand communities, live interactions, and sports communities, customers can establish social contact with enterprises or other customers, and engage in innovation activities(e.g., provide Feedback and Experience on products), thereby facilitating the development of fitness apps [8–11]. The present study aimed to uncover this mechanism in the context of fitness apps [6]. Moreover, studies have focused on knowledge management or flow experience but have not investigated their effects on customer value co-creation [13, 17]. In particular, customer knowledge management aids in value co-creation by improving customers' flow experience [14]. Therefore, this study proposed and tested a model based on customer-dominant (C-D) logic, in which customer knowledge management affects customer value co-creation through customer flow experience in fitness apps.

The following research questions were also explored:

RQ1. How are customer knowledge management, flow experience, and customer value co-creation interrelated in the context of fitness apps?

RQ2. Does flow experience mediate the relationship between customer knowledge management and customer value co-creation in the context of fitness apps?

This study tested the proposed model by analyzing questionnaire data from 450 fitness app customers through structural equation modeling on SPSS and AMOS software (S1 Dataset). The results validated the model, indicating the significant positive effects of customer knowledge management on flow experience, significant positive effects of flow experience on customer value co-creation, and partial mediating effects of flow experience on the association between customer knowledge management and customer value co-creation.

The contribution of this study is twofold. First, it extends value co-creation theory to the context of fitness apps; the results indicate the importance of value co-creation for the fitness apps of enterprises. Second, by demonstrating how these three factors are interrelated in the context of fitness apps, it bridges the literature gap on customer knowledge management, flow experience, and customer value co-creation.

## 2. Literature review

### 2.1 Value co-creation theory

Value co-creation theory has drawn considerable academic attention. This theory has evolved from being based on good-dominant (G-D) logic, where value is created only by the goods provided by businesses to customers [19]; to service-dominant (S-D) logic, where customer services are considered the main contributor to value and are the foundation of economic exchange and which emphasizes high levels of business–customer interactions than those in G-D logic [20]; to the current iteration of C-D logic, where value is derived from value co-creation, a flexible value network, and personalization of the goods and services provided [14].

In C-D logic, customers are the main source of value [14]. Moreover, customer value co-creation helps businesses to gain a more effective understanding of customers and to meet the heterogeneous needs of their customers (Fig 1) [21]. In the internet platform and brand community, through interactive, businesses can apply artificial intelligence and data analysis to obtain customer insights, or fitness online classes can be provided to meet the needs of specific groups of customers according to their level and goals [22]. For example, through the popular Chinese fitness app Keep, 21,200 online fitness classes were provided to 1.3 million users in December 2022. In particular, live fitness classes enable direct interactions between users and

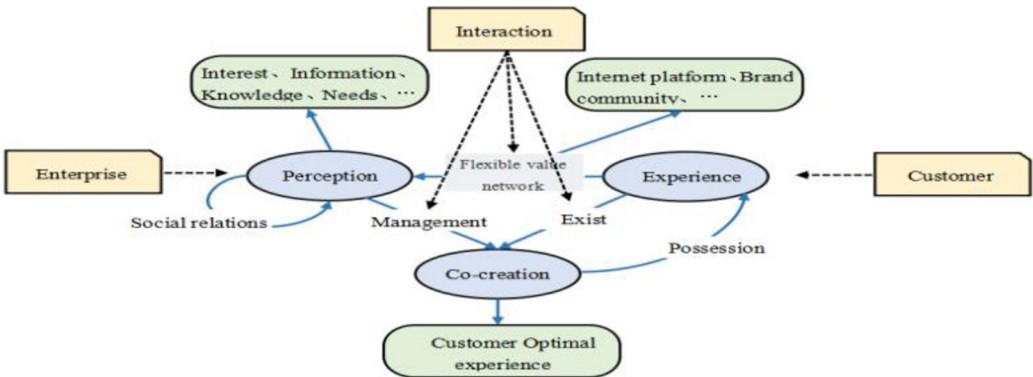

**Fig 1. Customer-dominant(C-D) logic.** This figure illustrates the relationship and process of value co-creation between customers and enterprises within the framework of customer - led logic.

the coach, for example, for the adjustment of the course content according to the user's needs. By 2022, Keep provided 9100 live online fitness classes to 17.1 million users. Therefore, understanding the crucial role of customer value co-creation and participatory process of customer value co-creation in the context of fitness apps is the major priority of fitness industry.

According to value co-creation theory in the context of C-D logic, customer knowledge management can be used to improve value co-creation by enhancing flow experience in fitness apps. This is exemplified by Keep. On this app, health consultants recommend exercise levels, fitness goals, everyday exercises, and dietary options tailored to users, enabling users to more effectively immerse themselves in the online fitness classes provided. The flow state then causes this user to engage with the app via a social feature where he recommends the specific exercise experience to other users. The Keep app leverages the knowledge shared by customers among themselves or with the company (i.e., customer knowledge management) for value co-creation, thus providing products and services that help customers achieve the flow state. This approach has resulted in 36.4 million active users and approximately 3.62 million subscribers monthly on the Keep app, with a membership penetration rate of approximately 10%. Thus, customer knowledge management, flow experience and value co-creation in C-D logic were considered in this study.

## 2.2 Customer value co-creation in fitness apps

Studies have categorized customer value co-creation into co-creation resulting from customer participation and co-creation resulting from customer citizenship [23], both of which are key in fitness apps (Table 1). Studies have demonstrated the ubiquity and importance of value co-creation in various industries with high customer participation, such as health care,

**Table 1. Customer value co-creation.**

| Value co-creation type | Dimensions | Definition in the context of fitness apps | Examples |
|---|---|---|---|
| Customer participation behavior | Information seeking, information sharing, responsible behavior, and personal interaction (Yi & Gong, 2011; Lee & Kim, 2021) | Active interaction with others, sharing of content among customers, and searching for information on novel fitness methods. | These novel fitness methods include fat loss, leg training, back training, or weight loss tips that customers share among themselves. |
| Customer citizenship behavior | Feedback, advocacy, helping, and tolerance (Yi & Gong,2011; Lee & Kim, 2021) | Active interaction with the company or other users to facilitate the more effective use of the company's products or services. | Customers may recommend other apps, share fitness methods, or give feedback on the company's products or services. |

tourism, and sports [24]. The customers of the sports industry have widely different personalities, are highly willing to participate, and are capable of drawing on a wide range of resources, thus facilitating value co-creation [25]. Previous studies have uncovered the effects of customer value co-creation for sports events spectators and participants in fitness activities, which feature high levels of business–customer and customer–customer interactions [26]. Thus, in the sports and fitness industry, the involvement of many actors in value co-creation improves customer satisfaction [27] and customer loyalty [28].

Customer knowledge management (CKM) is a process in which enterprises or service organizations manage customers' ideas, information, problems, needs and preferences for products or services [29]. Previous studies have highlighted that customer knowledge management crucially affects customer value co-creation [30]. Customer knowledge management not only can help enterprises and service organizations achieve their innovation and development goals but also can facilitate the establishment of co-creation relationships between such organizations and customers [16]. For example, Valacherry et al. used qualitative case studies to discover the relationship between customer knowledge management and customer value co-creation in retail contexts; the researchers discovered that customer knowledge management under the framework of social media can improve business-oriented relationships and promote active customer engagement, interaction, and the sharing of knowledge [31]. However, many managers take only simple marketing steps toward customer knowledge management and customer value co-creation and fail to consider the process through which customer value co-creation develops [32].

Csikszentmihalyi proposed the concept of the flow experience, which relates to an individual's psychological state during an interactive experience [33]. Flow experience refers to a complex psychological phenomenon that leads to enjoyment, a loss of self-awareness, and voluntary reinforcement [34]. Previous studies have highlighted that in online service contexts involving human–computer interaction, flow experience has been employed extensively to explain customer behaviors [35]. For example, in a study regarding factors influencing the loyalty of mobile instant messaging users, flow experience had a significant and positive impact on users' perceived satisfaction and loyalty [33–35]. In addition, Kim et al. used customer flow experience as a mediating variable related to the live streaming of esports and verified the mechanism of action between esports viewing motivation and esports fans' behavioral intentions and levels of video game loyalty [36].

A fitness app is a product involving online human–computer interaction; in such products, customer flow experience is simple to generate and has a crucial impact on customer behaviors [37]. Furthermore, customers can create value by sharing their experiences(like ideas, information, problems, needs etc.) in virtual sports communities or on fitness platforms [14, 15], Customer knowledge management has a positive effect on customer value co-creation. However, no study has investigated customer value co-creation in the context of fitness apps. Studies have investigated the outcomes of customer value co-creation in sports [26–28], such as customer satisfaction or loyalty. Given that they facilitate social media and multiagent interactions, fitness apps have transformed the generation of such co-creation. Thus, the present study explored the role of customer value co-creation in the development of fitness apps.

In brief, to compensate for the shortcomings of previous research, the present study considered variables such as customer knowledge management and flow experience when exploring the influencing factors and participatory process of customer value co-creation in the context of fitness apps; the objectives were to enrich the existing literature regarding customer value co-creation in sporting contexts and to provide a novel research perspective to aid in the development of fitness apps.

## 2.3 Theoretical and hypothesis development

**2.3.1 Customer knowledge management and flow experience.** Flow experience, as first conceptualized by Csikszentmihalyi, is a complex psychological phenomenon in which an individual is immersed in and completely focused on an activity [33]. Of the dimensions of flow experience conceptualized by scholars, previous studies have mostly focused on perceived enjoyment and attention focus. Perceived enjoyment indicates how much a customer experiences enjoyment and pleasure when using a product or service [38]. Attention focus refers to how much customers enjoy immersive experience when using a product or service [39]. Thus, in the present study, flow experience was measured in terms of these two dimensions [40, 41].

Through customer knowledge management, businesses can gain a comprehensive understanding of customers' perception of the quality of the service provided [42]. Thus, customer knowledge management can be applied to increase customer enjoyment and to cultivate stronger relationships between businesses and customers [43]. Such perceived enjoyment, as a dimension of flow experience, is thus likely to improve flow [38].

Attention focus is a key part of customer experience [39]. Moreover, using the insights on customer behavior gained through customer knowledge management, businesses can provide more immersive experiences, which promote customer participation [44]. For example, through customer knowledge management, fitness app developers can determine the fitness goals of customers and their preferred exercise methods and can improve their products or services accordingly. Thus, this study proposed the following hypotheses:

**H1:** Customer knowledge management significantly and positively affects customer flow experience.

H1a: Customer knowledge management significantly and positively affects customer perceived enjoyment.

H1b: Customer knowledge management significantly and positively affects customer attention focus.

**2.3.2 Flow experience and customer value co-creation.** Customer value co-creation encompasses two dimensions: customer participation behavior and customer citizenship behavior [23]. Customer participation behavior relates to customers' perceptions of the value of a product or service and is demonstrated through their levels of activity in searching for information, sharing knowledge, and participating in related interactions. Customer citizenship behavior refers to behaviors (e.g., feedback, support, and tolerance) that customers voluntarily engage in to increase the value of a product or service [11]. Some studies have reported that customers' perceived value is an antecedent of customer value co-creation [45]. When customers enjoy certain services and products and perceive quality in their own information, they become more enthusiastic and more willing to engage, leading them to more actively participate in value co-creation [46]. A customer's perceived enjoyment of flow experience represents their perceived value of a specific product or service [38]. Theoretically, a customer who enjoys engaging with a product or service views that product or service as a central element of their life and uses it in a manner that expresses their identity [22]; thus, they are invested in the product or service and likely to participate in value co-creation.

Attention focus with respect to experience reflects the degree of immersion experienced by a customer while they are using a product or service [39]. One study indicated a significant relationship between customer experience and attention focus related to flow experience [35]. For example, when a customer is using a fitness app, if they perceive a low degree of immersion, they may have an unsatisfactory experience. At the same time, in the value co-creation mechanism that satisfies C-D logic, customer experience serves as the basis for customer participation in value co-creation [14]. The more focused a customer's attention is, the stronger is

their feeling of immersion, the more positive is their customer experience, and the more active is their engagement in value co-creation (i.e., through customer participation behavior and customer citizenship behavior) [23]. In other words, customers actively participate in value co-creation when in an attention-focused state that is induced by an experience perceived to be positive. Accordingly, this study proposed the following hypotheses:

**H2:** Flow experience significantly and positively affects customer value co-creation.

H2a: Perceived enjoyment significantly and positively affects customer participation behavior.

H2b: Perceived enjoyment significantly and positively affects customer citizenship behavior.

H2c: Attention focus significantly and positively affects customer participation behavior.

H2d: Attention focus significantly and positively affects customer citizenship behavior.

**2.3.3 Mediating effects of flow experience.** According to H1 and H2, customer knowledge management contributes to customer value co-creation through flow experience. Statistically, mediation is demonstrated by a significant indirect influence [47]. According to consumption psychology and the social exchange theory of emotions, people are motivated to pursue positive emotional experiences, and emotions experienced during such experiences can be contagious [48]. The creation of contagious emotional experiences is affected by external stimulation and the surrounding environment [49]. Therefore, in customer value co-creation, customers are vulnerable to external stimulation from enterprises engaging in customer knowledge management, which can generate emotional experiences such as perceived enjoyment and attention focus. These emotional experiences influence customers' engagement in value co-creation, including their participation behavior (e.g., information searching, information sharing, and interpersonal interaction) and citizenship behavior (e.g., providing feedback, support, and assistance) in the consumption process. Accordingly, this study proposed the following hypotheses:

**H3:** Flow experience mediates the association between customer knowledge management and customer value co-creation.

H3a: Perceived enjoyment and attention focus mediate the effects of customer knowledge management on participation behavior.

H3b: Perceived enjoyment and attention focus mediate the effects of customer knowledge management on citizenship behavior.

## 2.4 Conceptual model

The aforementioned hypotheses were integrated into a model based on C-D logic, in which customer knowledge management (the independent variable) affects customer value co-creation (the dependent variable) through customer flow experience (the mediating variable) (Fig 2).

H1a and H1b pertained to the relationships of customer knowledge management with the perceived enjoyment and attention focus dimensions of flow experience, respectively. H2a, H2b, H2c, and H2d pertained to the relationships of the aforementioned two dimensions of flow experience with the participation and citizenship dimensions of customer value co-creation. Finally, H3a and H3b pertained to the mediating effects of flow experience on the relationship between customer knowledge management and customer value co-creation.

## 3. Methods

### 3.1 Study sample

Fitness apps in China are utilized for maintaining health records (e.g., *Gudong* and Yue Running Circle), or they provide assistance with maintaining or improving fitness (e.g., Keep,

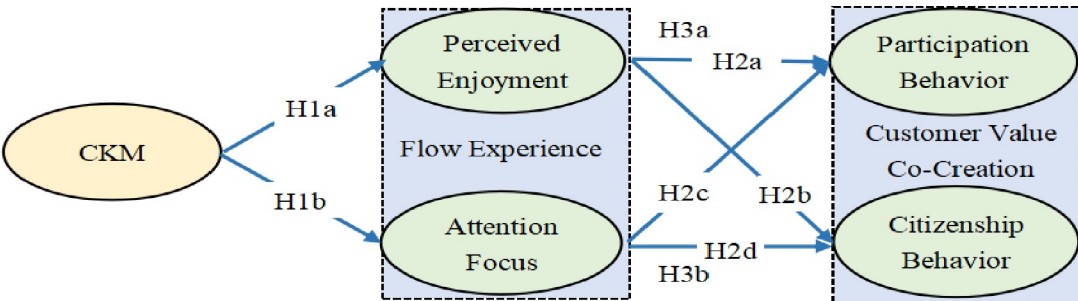

**Fig 2. Conceptual model.** This figure depicts the hypothesized influence paths. The independent variable is anticipated to impact the dependent variable via the mediating effect.

Millet Sports and BooHee Health). These apps also provide consultations regarding fitness or diet or are subscription services. This study focused on fitness apps applied for maintaining health records and providing fitness assistance because they feature frequent customer–customer and customer–business interactions [50] and they help customers achieve their goals and engage in co-creation [24]. To ensure the authenticity and reliability of the research sample, we introduced a screening question at the outset of the questionnaire specifically designed to identify these users. The study procedures were approved by the Medical Ethics Committee of Wuhan Sport University(2023101). Informed consent forms were signed and obtained from all individual participants included in the study.

## 3.2 Measurement of variables

This study adopted scales from the literature for measuring customer knowledge management, flow experience, and customer value co-creation. Three experts in sports management were invited to provide suggestions for modifying the scale items. All scale items were rated on a 7-point Likert scale (1 = *strongly disagree* and 7 = *strongly agree*).

Customer knowledge management was measured using the scale of Behnam et al (2020) [42]. As indicators of flow experience, perceived enjoyment and level of immersion were measured using the scale of Zhou and Lu (2011) [41]. As indicators of customer value co-creation, customer participation behavior and citizenship behavior were measured using the scale of Lee and Kim (2021) [11]. In this study, customer participation behavior was a second-order variable constituted by the dimensions of information searching, information sharing, responsibility behavior, and interpersonal interaction. Customer citizenship behavior was a second-order variable with the dimensions of feedback, support, assistance, and tolerance. The measurement of the variables are summarized in Tables 2 and 4.

## 3.3 Pilot study

To refine the questionnaire (S1 Appendix), a pilot study involving 30 respondents was conducted from October 1 to 15, 2021. All participants provided their written informed consent, and the data were anonymized and used only in the context of the present study. We only included respondents who used apps for maintaining health records and providing fitness assistance.

## 3.4 Research implementation

The formal survey was administered on the Questionnaire Star platform from October 17, 2021, to June 1, 2022. Convenience sampling was adopted to recruit the study sample.

**Table 2. Measurement of related variables.**

| Constructs | Dimensions | | Reference question | Number of items | References |
|---|---|---|---|---|---|
| Customer Value Co-creation | Customer participation behavior | Information seeking | e.g., "I have asked others for information on what this service offers, I have paid attention to how others behave to use this service". | 3 | Lee &Kim (2021) |
| | | Information sharing | e.g., "I clearly explained what I wanted the employee to do, I gave the employee proper information". | 4 | |
| | | Responsible behavior | e.g., "I performed all the tasks that are required, I fulfilled responsibilities to the business". | 4 | |
| | | Personal interaction. | e.g., "I was friendly to the employee, I was polite to the employee". | 4 | |
| | customer citizenship behavior | Feedback | e.g., "When I receive good service from the employee, I comment about it." | 3 | |
| | | Advocacy | e.g., "I am willing to say positive things about XYZ and the employee to others". | 3 | |
| | | Helping | e.g., "I help other customers if they seem to have problems". | 4 | |
| | | Tolerance | e.g., "If the employee makes a mistake during service delivery, I would be willing to be patient". | 3 | |
| Flow experience | Perceived enjoyment | | e.g., "I feel that using this mobile IM is fun, I feel that using this mobile IM is enjoyable." | 4 | Zhou & Lu (2011) |
| | Attention focus | | e.g., "When using this mobile IM, I was absorbed intensely in the Activity, When using this mobile IM, I concentrated fully on the activity." | 4 | |
| Customer knowledge management | | | e.g., "My club asks customers about its current service quality, My club demonstrates an understanding of its customer's background, My club provides information about current services for customers." | 5 | Behnam et al (2020) |

**Table 3. Sample demographics.**

| Classification indicator | Category | Frequency | Percentage (%) |
|---|---|---|---|
| Gender | Male | 219 | 49% |
| | Female | 231 | 51% |
| Age | 10–17 | 5 | 1.2% |
| | 18–25 | 420 | 93.3% |
| | 26–35 | 17 | 3.7% |
| | 36–45 | 4 | 0.9% |
| | >46 | 4 | 0.9% |
| Education | High school and below | 13 | 2.9% |
| | College | 15 | 3.3% |
| | Undergraduate | 304 | 67.6% |
| | Master's degree and above | 118 | 26.2% |
| Monthly income | <3000 | 371 | 82.5% |
| | 3001–5000 | 31 | 6.9% |
| | 5001–8000 | 20 | 4.4% |
| | >8000 | 28 | 6.2% |
| monthly usage | <3 | 219 | 48.6% |
| | 3–8 | 121 | 26.9% |
| | 9–15 | 53 | 11.8% |
| | >15 | 57 | 12.7% |

**Table 4. Questionnaire items.**

| VARIABLE/ITEMS | Λ | SE | AVE | CR | CRONBACH'SA |
|---|---|---|---|---|---|
| CUSTOMER KNOWLEDGE MANAGEMENT | | 0.423 | 0.9444 | 0.9884 | 0.988 |
| THE APP ASKS ME ABOUT WHAT ATHLETIC ACTIVITIES I AM INTERESTED IN AND HOW HIGHLY I RATE THE QUALITY OF SERVICE. | 0.980 | | | | |
| THE APP UNDERSTANDS MY NEEDS AND SOLVES MY PROBLEMS IN A TIMELY MANNER. | 0.972 | | | | |
| THE APP HAS A GOOD GRASP OF MY FITNESS PREFERENCES. | 0.964 | | | | |
| THE APP PROVIDES ME WITH UPDATES AND INFORMATION ON NEW PRODUCTS. | 0.975 | | | | |
| THE APP OFFERS SUGGESTIONS TO HELP ME MAKE BETTER CONSUMPTION DECISIONS. | 0.968 | | | | |
| PERCEIVED ENJOYMENT | | 0.191 | 0.915 | 0.9773 | 0.976 |
| IT'S INTERESTING TO USE THE APP. | 0.944 | | | | |
| IT'S EXCITING TO USE THE APP. | 0.972 | | | | |
| IT'S PLEASANT TO USE THE APP. | 0.960 | | | | |
| IT'S ENJOYABLE TO USE THE APP. | 0.950 | | | | |
| ATTENTION FOCUS | | 0.376 | 0.957 | 0.9889 | 0.988 |
| I AM HIGHLY ATTRACTED TO THE APP'S PRODUCTS AND SERVICES. | 0.980 | | | | |
| I AM IN A STATE OF DEEP FOCUS WHEN USING THE APP. | 0.976 | | | | |
| I AM IN A STATE OF DEEP FOCUS WHEN USING THE APP'S PRODUCTS AND SERVICES. | 0.978 | | | | |
| I GIVE MY ALL TO THE ATHLETIC ACTIVITIES ASSOCIATED WITH THE APP. | 0.979 | | | | |
| CUSTOMER PARTICIPATION BEHAVIOR | | | | | |
| INFORMATION SEEKING | | | | | |
| I ASK OTHERS ABOUT ATHLETIC PRODUCTS AND SERVICES. | 0.889 | 0.153 | 0.7953 | 0.921 | 0.918 |
| I HAVE SEARCHED FOR LOCATION INFORMATION ABOUT SPORTS PRODUCTS AND SERVICES | 0.910 | | | | |
| I HAVE OBSERVED HOW OTHER PEOPLE USE FITNESS APPS. | 0.876 | | | | |
| INFORMATION SHARING | | | | | |
| I HAVE CLEARLY EXPRESSED MY NEEDS WITH REGARD TO THE PRODUCTS OR SERVICES PROVIDED BY THE COMPANY. | 0.939 | 0.133 | 0.792 | 0.9383 | 0.936 |
| I HAVE PROVIDED USEFUL INFORMATION TO HELP THE COMPANY INNOVATE IN THEIR PRODUCTS AND SERVICES. | 0.873 | | | | |
| I HAVE OFFERED FEEDBACK ON HOW THE COMPANY CAN PROVIDE SERVICES IN A TIMELY MANNER. | 0.870 | | | | |
| I ANSWER ALL QUESTIONS POSED TO ME ABOUT SPORTS PRODUCTS AND SERVICES. | 0.876 | | | | |
| RESPONSIBLE BEHAVIOR | | | | | |
| I HAVE COMPLETED ALL THE STEPS REQUIRED BY THE APP. | 0.818 | 0.135 | 0.7178 | 0.9104 | 0.916 |
| I HAVE UNDERTAKEN ALL ACTIONS REQUIRED BY THE APP. | 0.819 | | | | |
| I HAVE FULFILLED ALL TASKS REQUIRED BY THE APP. | 0.875 | | | | |
| I HAVE COMPLIED WITH ALL INSTRUCTIONS GIVEN BY THE APP. | 0.875 | | | | |
| PERSONAL INTERACTION | | | | | |
| I AM FRIENDLY TO CUSTOMER SERVICE STAFF ONLINE AND OFFLINE. | 0.945 | 0.112 | 0.8563 | 0.9597 | 0.959 |
| I AM KIND TO CUSTOMER SERVICE STAFF ONLINE AND OFFLINE. | 0.903 | | | | |
| I AM POLITE TO CUSTOMER SERVICE STAFF ONLINE AND OFFLINE. | 0.924 | | | | |
| I AM COURTEOUS TO CUSTOMER SERVICE STAFF ONLINE AND OFFLINE. | 0.929 | | | | |
| I AM FRIENDLY TO CUSTOMER SERVICE STAFF ONLINE AND OFFLINE. | | | | | |
| CUSTOMER CITIZENSHIP BEHAVIOR | | | | | |
| FEEDBACK | | | | | |
| I LET THE COMPANY KNOW WHEN I HAVE ANY FEEDBACK. | 0.886 | 0.125 | 0.8151 | 0.9297 | 0.930 |
| I AM WILLING TO COMMENT ON CONTENT ON THE APP WHENEVER I ENCOUNTER IT. | 0.896 | | | | |
| I LET THE COMPANY KNOW WHEN I ENCOUNTER PROBLEMS IN USING THE APP. | 0.926 | | | | |
| ADVOCACY | | | | | |

*(Continued)*

**Table 4.** (Continued)

| VARIABLE/ITEMS | Λ | SE | AVE | CR | CRONBACH'SA |
|---|---|---|---|---|---|
| I WILL GIVE THE FITNESS APP A HIGH RATING ON THE APP STORE. | 0.869 | 0.110 | 0.7646 | 0.9069 | 0.907 |
| I WILL RECOMMEND THE FITNESS APP TO OTHERS. | 0.861 | | | | |
| I WILL RECOMMEND THE FITNESS APP TO MY FRIENDS AND RELATIVES. | 0.870 | | | | |
| HELPING | | | | | |
| I HELP OTHER APP USERS IN NEED. | 0.870 | 0.109 | 0.8095 | 0.9444 | 0.938 |
| I HELP OTHER APP USERS IN USING THE APP. | 0.923 | | | | |
| I HELP OTHER APP USERS USE THE APP CORRECTLY. | 0.901 | | | | |
| I GIVE ADVICE TO OTHER APP USERS. | 0.904 | | | | |
| TOLERANCE | | | | | |
| I STILL ACCEPT THE APP EVEN IF IT DOES NOT WORK AS EXPECTED. | 0.897 | 0.166 | 0.7712 | 0.910 | 0.913 |
| I WORK AROUND FAILURES IN THE FITNESS APP. | 0.880 | | | | |
| I WAIT PATIENTLY EVEN IF THE APP TAKES A LONG TIME TO LOAD. | 0.857 | | | | |

Respondents who completed the questionnaire were from college classes or fitness groups. Most respondents were from Fujian, Hubei, or Henan Province.

In total, 450 valid questionnaires of 550 questionnaires were collected. The questionnaires that were completed too quickly or had failed consistency checks (including those with highly consistent scores for different latent variables) were excluded. The number of valid questionnaires (n = 450) was 10 times higher than the number of items in questionnaire (41 analysis items); this sample size met the requirements for structural equation modeling [51]. The questionnaire recovery rate and efficiency rate were 94% and 87%, respectively. The demographic characteristics of the respondents are summarized in Table 3. Most respondents (93.3%) were aged 18 to 25 years, followed by the age group of 26 to 35 years (3.7%). This distribution was reflective of the age distribution of the fitness app–using population in China, as reported in the China Mobile Fitness Industry Report released by iResearch Consulting in 2021 [50].

## 4. Results

### 4.1 Measurement model

All statistical analyses were conducted using AMOS version 24.0 or SPSS version 23.0 software. The psychometric properties of the questionnaire were first tested. The results of confirmatory factor analysis indicated that the measurement models had good fit ($\chi^2$/degrees of freedom [df] = 1.569, root mean square residual = 0.063, GFI = 0.902, AGFI = 0.865, TLI = 0.981, CFI = 0.986, RMSEA = 0.036).

The customer knowledge management, flow experience, and customer value co-creation scales also had high composite reliability and internal consistency. Specifically, their composite reliability values (0.910 to 0.989) and Cronbach's alpha values (0.907 to 0.988) were greater than the requisite minimum of 0.7 [12].

The scales had content validity because they were adopted from scales in the literature [11, 41, 42] and were modified based on suggestions from three experts [3, 52].

The scales also had construct validity. Specifically, the scales had convergent validity, as (1) the standardized factor loadings of all items were significant and greater than 0.6 [3], (2) the average variance extracted (AVE) was greater than 0.5 [11], and (3) the combined reliability was greater than 0.7 [42] (Table 4). The latent variables also had discriminant validity, as the square root of the AVE of each latent variable was greater than the correlation coefficients between that variable and the other variables [41] (Table 5).

**Table 5. Correlation coefficient of latent variables.**

|      | To    | He    | Ad    | Fe    | PI    | RB    | IS    | ISe   | AF    | PE    | CKM   |
|------|-------|-------|-------|-------|-------|-------|-------|-------|-------|-------|-------|
| To   | 0.878 |       |       |       |       |       |       |       |       |       |       |
| He   | 0.806 | 0.900 |       |       |       |       |       |       |       |       |       |
| Ad   | 0.822 | 0.829 | 0.874 |       |       |       |       |       |       |       |       |
| Fe   | 0.841 | 0.849 | 0.828 | 0.903 |       |       |       |       |       |       |       |
| PI   | 0.686 | 0.796 | 0.873 | 0.771 | 0.925 |       |       |       |       |       |       |
| RB   | 0.833 | 0.817 | 0.843 | 0.815 | 0.818 | 0.847 |       |       |       |       |       |
| IS   | 0.753 | 0.764 | 0.812 | 0.824 | 0.731 | 0.826 | 0.890 |       |       |       |       |
| ISe  | 0.758 | 0.783 | 0.784 | 0.806 | 0.693 | 0.826 | 0.855 | 0.892 |       |       |       |
| AF   | 0.614 | 0.608 | 0.618 | 0.629 | 0.562 | 0.658 | 0.664 | 0.749 | 0.978 |       |       |
| PE   | 0.582 | 0.576 | 0.651 | 0.612 | 0.592 | 0.662 | 0.652 | 0.671 | 0.586 | 0.957 |       |
| CKM  | 0.405 | 0.528 | 0.571 | 0.477 | 0.547 | 0.544 | 0.503 | 0.489 | 0.430 | 0.526 | 0.972 |

Note(s):To = tolerance; He = helping; Ad = advocacy; Fe = feedback; PI = personal interaction; RB = responsible behavior; IS = information sharing; ISe = information seeking; AF = attention focus; PE = perceived enjoyment; CKM = customer knowledge management; Data on the diagonal are square roots (average variance extracted).

## 4.2 Common method bias

In this study, procedural control was used to minimize common method bias given that all data were collected using a single questionnaire [53], and statistical control was then applied to test for common method bias.

Specifically, in this study, the surveys were anonymized, the order of questions disrupted, and guides were added to highlight key content for reducing common method bias. No major common method bias was detected; a Harman's single factor test revealed a common factor interpretation rate, which was less than the requisite maximum of 40%.

## 4.3 Structural modeling

The structural model had good fit ($\chi^2$/df = 2.325, GFI = 0.946, AGFI = 0.921, TLI = 0.960, CFI = 0.964, RMSEA = 0.054).

The results (Fig 3) of structural equation modeling revealed that customer knowledge management significantly and positively influenced the perceived enjoyment ($\gamma$ = 0.525, P < 0.001) and attention focus ($\gamma$ = 0.424, P < 0.001) dimensions of flow experience. H1a and H1b were supported by the findings of bootstrap analysis. These findings are exemplified by the services provided by Nike Training Club app. Data analytics of track-running records stored on the Nike Training Club app are conducted to recommend services (e.g., targeted training programs), ensuring a more enjoyable and attention focus experience for the customer.

Perceived enjoyment and attention focus significantly and positively influenced the participation ($\gamma$ = 0.470, P < 0.001 and $\gamma$ = 0.431, P < 0.001, respectively) and citizenship ($\gamma$ = 0.410, P < 0.001 and $\gamma$ = 0.418, P < 0.001, respectively) dimensions of customer value co-creation. H2a–H2d were supported by the findings of bootstrap analysis. The findings are exemplified by the Peloton fitness app. Live and on-demand training classes on the Peloton fitness app can be viewed offline to enhance the user's perceived enjoyment of and attention focus in the training classes, in turn promoting value co-creation through user interactions.

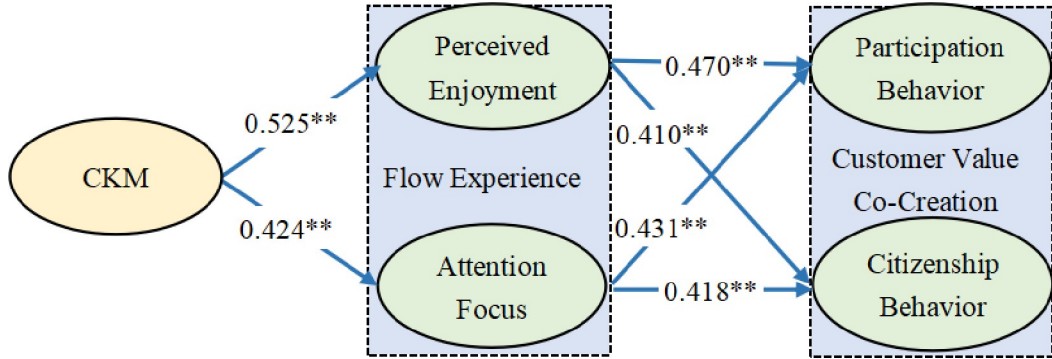

**Fig 3. Path analysis results.** The figure shows standardized path coefficients of influence paths in the structural equation model. ** indicates P < 0.01, denoting a highly significant relationship.

## 4.4 Mediating effect testing

The aforementioned mediating effect of flow experience was tested (Table 6). Bootstrap sampling was conducted with 5000 bootstrap samples. The nonparametric percentile method was used for bias correction. Moreover, 95% confidence intervals (CIs) were estimated.

The effects of customer knowledge management on the participation and citizenship dimensions of co-creation were significantly mediated by perceived enjoyment (95% CI: 0.052 to 0.117 and 0.111 to 0.112, respectively) and immersion (95% CI: 0.059 to 0.108 and 0.033 to 0.070, respectively). Thus, H3a and H3b were supported.

A further analysis using an intermediary model was conducted. The results revealed that customer knowledge management had a significant direct effect on customer participation behavior (0.087; 95% CI: 0.055 to 0.122) and customer citizenship behavior (0.117; 95% CI: 0.077 to 0.162). Perceived enjoyment and attention focus partially mediated the effects of customer knowledge management on customer participation and citizenship behavior.

## 5. Discussion

This study is the first to demonstrate that customer knowledge management influences customer value co-creation (in terms of participation and citizen behaviors) through flow experience (in terms of attention focus and perceived enjoyment) in fitness apps.

First, the results of this study indicate that customer knowledge management in relation to fitness apps can affect customers' flow experiences. Customer knowledge management had a

**Table 6. Mediating effect testing results.**

| Path | Effect | Boot SE | P | Bias-Corrected 95% CI | | Percentile 95% CI | |
|---|---|---|---|---|---|---|---|
| | | | | Lower | Upper | Lower | Upper |
| CKM→PE→CPB | 0.081 | 0.017 | 0.000 | 0.052 | 0.117 | 0.056 | 0.110 |
| CKM→AF→CPB | 0.081 | 0.012 | 0.000 | 0.059 | 0.108 | 0.059 | 0.099 |
| CKM→PE→CCB | 0.155 | 0.026 | 0.000 | 0.111 | 0.211 | 0.115 | 0.199 |
| CKM→AF→CCB | 0.049 | 0.009 | 0.000 | 0.033 | 0.070 | 0.034 | 0.065 |
| CKM→CPB | 0.087 | 0.017 | 0.000 | 0.055 | 0.122 | 0.059 | 0.116 |
| CKM→CCB | 0.117 | 0.022 | 0.000 | 0.077 | 0.162 | 0.081 | 0.151 |

Note(s); CKM = customer knowledge management; PE = perceived enjoyment; AF = attention focus; CPB = customer participation behavior; CCB = customer citizenship behavior; Relevant Numbers in the Table are Non-standardized Values

significant positive impact on the two dimensions of flow experience, namely, perceived enjoyment and attention focus. The results of this study are consistent with those in the literature regarding customer knowledge management. Specifically, previous studies have reported that customer knowledge management can create value for enterprises and service organizations by enhancing customer experience [54]. Customer knowledge management serves as the foundation for customers' perceived experience; therefore, customer knowledge management can improve the quality of services through the acquisition, utilization, and integration of customer knowledge, each of which has a crucial effect on customers' cognition during consumption [55]. In summary, the findings of the present study support those of previous knowledge management research.

In contrast to previous studies, the results of the present study support the relationships of customer knowledge management with both dimensions of flow experience, namely, perceived enjoyment and attention focus, in the context of fitness apps. A previous study regarding flow experience indicated that to ensure the generation of perceived enjoyment and attention focus during flow experience, enterprises must understand customers' skills and challenges [33]. When an appropriate balance between a customer's skill level and the challenges that they are assigned is struck, perceived enjoyment and attention focus are generated. When their skill level is greater than that required by a challenge, the customer feels bored; similarly, a customer feels anxious when the level of a challenge exceeds their current skill level [41]. A customer's skill level with respect to the challenges they can comfortably face constitutes customer knowledge [56]. Thus, striking the appropriate balance requires the comprehensive identification and management of customer knowledge. For example, a fitness app can determine a customer's physical fitness through customer knowledge management and accordingly develop tailored training courses for that customer. In this way, the app recognizes the required balance between skill and challenge and ultimately enables the customer to have a flow experience with perceived enjoyment and attention focus.

Second, the results of this study indicate that flow experience can affect customer value co-creation in the context of fitness apps. Customer flow experience involving perceived enjoyment and attention focus was found to be positively related to customer value co-creation in the form of both customer participation behavior and citizenship behavior. This finding empirically support claims made in the literature. Studies have demonstrated the impact of customers' flow experiences on their consumption behaviors, including their behaviors related to loyalty and satisfaction [35, 36]. In addition, research has also indicated that customers perceive extremely high levels of enjoyment during a flow experience [33]. In such a state, creativity and creative behaviors are greatly mobilized [57]. For example, Primus and Sonnenburg identified flow experience as a crucial precursor to frequent creative and innovative behaviors in the splicing process of Lego games [58]. Studies have shown that customer value co-creation not only is a key consumption behavior but also constitutes a creative activity [11]. By demonstrating flow experience to be an essential customer consumption behavior and creative activity, the results of present study empirically support the relationship between flow experience and customer value co-creation.

In contrast to the aforementioned research, the results of the present study indicate that a high level of flow experience (involving perceived enjoyment and attention focus) makes fitness app customers more willing to engage in value co-creation and citizenship behavior because perceived enjoyment and attention focus are personalized products created in the context [41]. Research has shown that sporting activities featured on fitness apps synergize with interactivity to cultivate perceived enjoyment and attention focus in app users [3, 41]. A customer's levels of perceived enjoyment and attention focus during while using a fitness app strengthen the relationship between the fitness app and the user and stimulate curiosity and

desire for exploration [33]. For example, the recommendation and sharing of sports equipment in fitness apps, the willful communication of sports experiences, the team challenge of community activities, and the mutual help with and solution of difficult problems. That is, fitness apps bring pleasure and high levels of perceived enjoyment and attention focus to their users, motivating these users to further engage in value co-creation.

Third, the results of this study indicate that flow experience plays a partial mediating role in the relationship between customer knowledge management and customer value co-creation. Specifically, perceived enjoyment and attention focus play partial mediating roles in the effects of customer knowledge management on customer value co-creation citizenship behavior and participation behavior. The mediating role of flow experience has been demonstrated in other digital service scenarios [36]. For example, Liu et al. demonstrated that customer flow experience mediates the interrelationships between interactivity, authenticity, and entertainment in live tourism e-commerce broadcasts and can influence the purchase intention of consumers watching such broadcasts [59]. Similarly, the present study observed a mediating role of flow experience.

In addition, this study's results support the impact of customer knowledge management on the two dimensions of flow experience, namely, perceived enjoyment and attention focus, as well as the impacts of both perceived enjoyment and attention focus on customer participation behavior and citizenship behavior in the context of fitness apps. Social exchange theory holds that when individuals gain emotional benefits from an organization, they form a psychological connection with that organization [48]. In the context of fitness apps, customers can share knowledge, such as their experiences related to sports and fitness, through interactive channels such as virtual sports brand communities. Furthermore, enterprises and service organizations can optimize customers' perceived enjoyment and attention focus while consuming sports products or services through knowledge management. Finally, it will stimulate the common consciousness and emotional experience of the members, deepen the emotional connection and form the emotional community. Simultaneously, however, this sense of community and emotional connection have a migration effect [49]. As customer involvement in fitness apps increases, customers increasingly feel a sense of trust in the app and thus increasingly provide valuable feedback for fitness app developers; such feedback can help to motivate other app users to engage in co-creation and citizenship behaviors.

## 5.1 Implications of the study

**5.1.1 Practical implications.**   Customer knowledge management is key for fitness app enterprises. The insights obtained through customer knowledge management can be used to enhance customer flow experience, thus promoting customer value co-creation for app development.

First, fitness apps must improve customer knowledge management approaches to promote value co-creation. The content generated through co-creation provides a large amount of data; such data can be leveraged through customer knowledge management to provide training programs tailored to users. In particular, integrating instant feedback and reward functions into fitness apps should be prioritized to promote user interactions.

Second, fitness apps must promote the customer flow experience to strengthen the psychological connection of customers to the enterprise. In the experience economy era, fitness app users can choose products that match their personal preferences. Therefore, to improve customers' perception of fitness products and services, enterprises can mine data on customer characteristics to create exclusive, customized, or modifiable fitness products and services.

Enterprises can also expand service scenarios and improve product distribution or after-sales services to enhance customer experience.

Third, fitness apps can promote customer value co-creation and strengthen their relationship with customers. For this purpose, enterprises must continue to innovate in how they connect with customers and must obtain insights of the target market to effectively generate customer value co-creation. Innovative interactive communication could replace traditional user interaction in virtual brand communities and live interactions and customer functions, and permissions can be expanded to promote customer browsing, participation, and feedback reporting. Furthermore, reward mechanisms can be integrated into user interfaces to stimulate co-creation.

**5.1.2 Theoretical implications.**   First, this study introduced value co-creation theory into the literature on fitness apps to provide a novel research perspective and fresh research content to serve as a reference for the development of fitness apps and for further research regarding customer value co-creation within sporting contexts. Fitness apps are an effective platform for users to engage in customer value co-creation [14]. Specifically, customers can participate in value co-creation activities together with enterprises or service organizations through interactive channels such as virtual sports communities. However, no literature regarding the development of fitness apps has discussed the impact or process of value co-creation within the context of such apps [3–5]. In sporting contexts, research on customer value co-creation has often explored the outcomes of customer value co-creation [26–28]. However, the factors that influence customer value co-creation are unclear. Therefore, by using the value co-creation theory and C-D logic, the present study analyzed customer value co-creation in the context of fitness apps to provide not only a novel theoretical perspective regarding the development of fitness apps by enterprises and service organizations but also fresh research content related to customer value co-creation in sporting contexts.

Second, the present study explored the interrelationships between customer knowledge management, flow experience, and customer value co-creation to not only expand the scope of research related to customer knowledge management and flow experience but also enrich the research regarding the influence and process of customer value co-creation in the context of fitness apps. The relationship between customer knowledge management and customer value co-creation has been demonstrated in previous studies [30, 31]. However, no such process between customer knowledge management and customer value co-creation has been verified. In addition, although flow experience affects customer consumption behaviors during human–computer interaction [35, 36], no studies have investigated the impact of flow experience in fitness app service scenarios. The present study explored the relationship between customer knowledge management and customer value co-creation in the context of fitness apps and the mediating role of flow experience in this relationship. The findings of this study not only enrich the current body of research related to customer knowledge management and flow experience but also promote understanding of the factors that affect customer value co-creation in the context of fitness apps.

# 6. Conclusion and future directions

## 6.1 Conclusion

This study intends to examine the impact of customer knowledge management, flow experience and customer value co-creation and the mediating role of flow experience in the fitness app service scenario. From the findings of quantitative analysis, the following conclusions were drawn: First, customer knowledge management has a significant positive impact on customers' flow experience in the fitness app service scenario. Second, customers' flow experience

has a significant positive impact on customer value co-creation in the fitness app service scenario. Third, customers' flow experience plays a partial mediating role in the mechanism of customer knowledge management's effect on customer value co-creation in the fitness app service scenario.

## 6.2 Future directions

This study has several limitations. First, the proposed model may not be generalized to other digital sports services. Second, we did not consider flow experience dimensions other than perceived enjoyment and immersion or customer value co-creation dimensions other than beyond participation and citizenship. Some researchers have distinguished between bottom-up (i.e., customer-initiated) value co-creation versus top-down (i.e., business-initiated) value co-creation, and future studies can consider this distinction in their analysis. Third, our analysis did not consider the effects of moderating variables.

## Supporting information

**S1 Dataset.**
(CSV)

**S1 Appendix. Survey questionnaire.**
(PDF)

## Author Contributions

**Conceptualization:** Zhigang Wang.

**Funding acquisition:** Kai Hua.

**Investigation:** Liandi Liu.

**Writing – original draft:** Chao Wang.

**Writing – review & editing:** Zhigang Wang, Kai Hua.

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
