## [Decision Letter · Decision Letter 0]

9 Jun 2024

PONE-D-24-07863Relationship between Customer Knowledge Management and the Value Co-Creation of Fitness Application Customers: Mediating Role of Flow ExperiencePLOS ONE

Dear Dr. wang,

Thank you for submitting your manuscript to PLOS ONE. After careful consideration, we feel that it has merit but does not fully meet PLOS ONE’s publication criteria as it currently stands. Therefore, we invite you to submit a revised version of the manuscript that addresses the points raised during the review process.

The paper investigates CKM, flow, and value co-creation in fitness apps. While well-organized, it's overly lengthy. It lacks clarity on why only fitness apps are studied and how flow is induced. Empirical reporting needs clarification on participant scenarios and temporal sequencing. Writing inconsistencies and grammatical errors persist, diminishing its publication suitability. Consequently, regretfully, the paper cannot be accepted for publication in its present state.

We look forward to receiving your revised manuscript.

Kind regards,

Sherlyn Villate

Support Staff - Lead

PLOS ONE

Reviewers' comments:

Reviewer's Responses to Questions

**Comments to the Author**

1. Is the manuscript technically sound, and do the data support the conclusions?

Reviewer #1: Yes

Reviewer #2: No

2. Has the statistical analysis been performed appropriately and rigorously? 

Reviewer #1: Yes

Reviewer #2: No

3. Have the authors made all data underlying the findings in their manuscript fully available?

Reviewer #1: Yes

Reviewer #2: No

4. Is the manuscript presented in an intelligible fashion and written in standard English?

Reviewer #1: Yes

Reviewer #2: No

5. Review Comments to the Author

Reviewer #1: The paper studies the effect of customer knowledge management and flow experience on customer value co-creation and the mediating role of flow experience in the context of fitness apps. The authors collected data using survey and conducted structural equation modeling (SEM) to test the hypotheses. The topic of the paper is very interesting. Here are my suggestions:

1. H1 proposes a positive effect. However, H1a and H1b do not have a direction. I would recommend add a direction to H1a and H1b hypotheses. The same logic applies H2.

2. Common method bias. The paper does not address the common method bias. I would recommend address the issue using statistical test.

Overall, the paper is of good quality. Good luck with the interesting project.

Reviewer #2: Review for: Relationship Between Customer Knowledge Management and the Value Co Creation of Fitness Application Customers: Mediating Role of Flow Experience.

Background: This paper includes a study examining the relationship between customer knowledge management (CKM), flow, and value co-creation within the domain of fitness applications. The paper studies an interesting topic, using a set of well-established constructs from the literature. I enjoyed reading the paper and learning about the topics. However, there is a vast area for improvement for the paper as it currently stands. I’ll focus my review on some of the critiques and how they could potentially be addressed.

Streamline the Introduction and Theory sections

The main critique I have is that the paper is far too long. There is only one study, yet the paper is around 35 pages. The organization by section is nice on the one hand. On the other hand, it is unclear to the reader why there needs to be an introduction, theoretical background, and hypothesis development sections. At the very least, I’d suggest combining the theoretical background and hypothesis development sections.

Improve the Concreteness of the Idea

For the background/introduction of the paper, it is a bit too general as it currently reads. Even after finishing the paper, I was left wondering why the study examines only fitness apps. Why (or why not) would the theoretical chain between CKM, Flow, and Co-Creation only exist for fitness apps? The authors note that fitness apps are popular in China and globally, which I agree with. But the decision to exclusively center the paper on the mobile fitness industry could be defended stronger. It is also worth noting (you mention this in section 4.3) that there are thousands of different fitness apps, meaning that there are endless different specific features, user interfaces, and recommendation systems that will vary depending on which specific app a participant is thinking about in your study. Value co-creation is also a large umbrella term. Why not just study rating of the app or word-of-mouth intentions?

Another question I had while reading the paper was how exactly flow is being induced. The key theoretical concepts are explained (although I would suggest moving the explanations to be earlier on in the paper), yet I had a tough time understanding exactly what is going through a consumer’s mind and in their behavior. Providing a specific example on the first page describing exactly what features of CKM can induce a flow state in a customer, which then leads this customer to co-creation would be extremely beneficial in my opinion. For example, I did not understand how a flow state is being elicited (is the customer experiencing a flow state from using the app itself, such as logging a run OR is the flow state induced from the exercise itself, such as going for the run). If you could give an example along the lines of, “the Nike training app suggests a 3 mile run to the user based on his data. The user then completes the run and experiences a flow state while doing so. The flow state then causes this user to engage with the app via a social feature where he recommends the specific run route to other users in his area”. I’m not sure if this is how you picture the effect occurring in real life, but if so it would be beneficial to spell out.

Empirical Model and Study Reporting

Figure 2 is helpful in outlining exactly what your hypotheses are and how everything is connected in your conceptual model. The convenience sampling method used is not ideal, especially since the ample appears to be almost entirely 18-25 year olds who use fitness apps. If available, why not use an online participant recruitment platform like Prolific or Amazon Mechanical Turk? Table 2 is helpful, but I would suggest spending more time explaining the backdrop of the study. The questions/concepts are somewhat clear, but I was unsure what specific scenario or fitness app your participants were thinking about/responding to when answering these questions. Did you ask participants to download a specific app? Or are these questions in reference to an app they’ve already downloaded? Or are you just asking the questions more abstractly?

Again, I like the model you outline and report in Figure 3. The question I have here is whether you can explain why flow comes before participation behavior. This point may not be valid if you had explained the study design better, but since I am confused I’ll ask if it is possible for flow experience to happen simultaneously during customer value co-creation. I’m skeptical you can claim mediation here since what you have is not actually an experiment with a manipulated independent variable preceding the dependent variable. I’d suggest an experiment where you experimentally manipulate CKM, then measure flow/attention, and then measure value co-creation.

Improvements in Writing

There are also many instances of typos/inconsistencies/grammatical errors in the manuscript which made it difficult to interpret. The word “APP” is arbitrarily capitalized in some cases but not others. There are hanging spaces in some areas whereas others need a space after a period. A few sentences were not complete. It could also be important to create a stronger hook for the importance of the finding. I’d imagine all fitness app designers strive to delight their customers, leading to enjoyment of their app and thus wanting to co-create. However, what could a designer specifically take away from the survey to specifically enhance their app?

This is clearly an important area of research. I wish the authors the best of luck as they continue to study this timely, interesting, and important topic.

6. PLOS authors have the option to publish the peer review history of their article (what does this mean?). If published, this will include your full peer review and any attached files.

Reviewer #1: No

Reviewer #2: No

---

## [Author Response · Author response to Decision Letter 0]

27 Aug 2024

Response to the editor

Comment: The paper investigates CKM, flow, and value co-creation in fitness apps. While well-organized, it's overly lengthy. It lacks clarity on why only fitness apps are studied and how flow is induced. Empirical reporting needs clarification on participant scenarios and temporal sequencing. Writing inconsistencies and grammatical errors persist, diminishing its publication suitability. Consequently, regretfully, the paper cannot be accepted for publication in its present state.

Response: Dear editor, thank you very much for your comments. According to your suggestions, the newly revised manuscript has made the following revisions :

1.The newly revised manuscript shortens the length.

2.About the reason why the fitness APP was studied, I made a demonstration in the introduction, 2.1 and 2.2 chapters of the manuscript.

3.About the reason why the variable of flow experience is included in the study, I cited the case in the introduction part and the value co-creation theory part of the manuscript.

4.In the empirical research section of the third chapter of the manuscript, the interviewees participation situation and the time of participation in the survey have been corrected.

5.The problem of consistent writing of ' APP ' in the manuscript has been corrected. And the grammatical errors in the manuscript have been corrected.

Response to the 1st Reviewer

Comment:The paper studies the effect of customer knowledge management and flow experience on customer value co-creation and the mediating role of flow experience in the context of fitness apps. The authors collected data using survey and conducted structural equation modeling (SEM) to test the hypotheses. The topic of the paper is very interesting. Here are my suggestions:

1. H1 proposes a positive effect. However, H1a and H1b do not have a direction. I would recommend add a direction to H1a and H1b hypotheses. The same logic applies H2.

2. Common method bias. The paper does not address the common method bias. I would recommend address the issue using statistical test.

Response: Dear reviewer 1, Thank you very much for reviewing the manuscript. According to your review of the manuscript, I made the following modifications in the revised manuscript :

1.The relationship or direction of the influence between the hypotheses has been modified.

2.The common method bias test has been supplemented in the fourth chapter of the manuscript.

Response to the 2st Reviewer

Comment:This paper includes a study examining the relationship between customer knowledge management (CKM), flow, and value co-creation within the domain of fitness applications. The paper studies an interesting topic, using a set of well-established constructs from the literature. I enjoyed reading the paper and learning about the topics. However, there is a vast area for improvement for the paper as it currently stands. I’ll focus my review on some of the critiques and how they could potentially be addressed.

Response: Dear reviewer 2, thank you very much for your review and recognition of the manuscript. Thank you very much for the review proposal. Next, I will modify and reply one by one according to your suggestions. Thank you.

Streamline the Introduction and Theory sections

The main critique I have is that the paper is far too long. There is only one study, yet the paper is around 35 pages. The organization by section is nice on the one hand. On the other hand, it is unclear to the reader why there needs to be an introduction, theoretical background, and hypothesis development sections. At the very least, I’d suggest combining the theoretical background and hypothesis development sections.

Response: The introduction and theoretical basis have been deleted. Specifically, the introduction deletes some meaningless paragraphs and optimizes the language of some paragraphs. The theoretical basis part combines the theoretical background and the hypothesis part.

Improve the Concreteness of the Idea

For the background/introduction of the paper, it is a bit too general as it currently reads. Even after finishing the paper, I was left wondering why the study examines only fitness apps. Why (or why not) would the theoretical chain between CKM, Flow, and Co-Creation only exist for fitness apps? The authors note that fitness apps are popular in China and globally, which I agree with. But the decision to exclusively center the paper on the mobile fitness industry could be defended stronger. It is also worth noting (you mention this in section 4.3) that there are thousands of different fitness apps, meaning that there are endless different specific features, user interfaces, and recommendation systems that will vary depending on which specific app a participant is thinking about in your study. Value co-creation is also a large umbrella term. Why not just study rating of the app or word-of-mouth intentions?

Response: In the manuscript, I mainly revised the two parts of theory and practice on why to study the concept of fitness app. First of all, in the theoretical part, through combing the influence relationship of each variable in the introduction, it points out the research questions in the study of fitness app customer value co-creation. At the same time, the literature review emphasizes this research gap and highlights the importance of studying fitness app customer value co-creation.

Secondly, in the part of practical demonstration, in the relevant theoretical basis, by referring to the case and application data of keep and combining with the theoretical basis, the manuscript once again demonstrates why the customer value co-creation of fitness app is explored. At the same time, based on the value co-creation theory of customer-led logic and related case data, it also demonstrates the relationship between knowledge management and flow experience and customer value co-creation. 

Another question I had while reading the paper was how exactly flow is being induced. The key theoretical concepts are explained (although I would suggest moving the explanations to be earlier on in the paper), yet I had a tough time understanding exactly what is going through a consumer’s mind and in their behavior. Providing a specific example on the first page describing exactly what features of CKM can induce a flow state in a customer, which then leads this customer to co-creation would be extremely beneficial in my opinion. For example, I did not understand how a flow state is being elicited (is the customer experiencing a flow state from using the app itself, such as logging a run OR is the flow state induced from the exercise itself, such as going for the run). If you could give an example along the lines of, “the Nike training app suggests a 3 mile run to the user based on his data. The user then completes the run and experiences a flow state while doing so. The flow state then causes this user to engage with the app via a social feature where he recommends the specific run route to other users in his area”. I’m not sure if this is how you picture the effect occurring in real life, but if so it would be beneficial to spell out.

Response: The reason why the flow experience was introduced has been clarified in the manuscript. At the same time, some practical cases are cited on the introduction and the literature review section.

Empirical Model and Study Reporting

Figure 2 is helpful in outlining exactly what your hypotheses are and how everything is connected in your conceptual model. The convenience sampling method used is not ideal, especially since the ample appears to be almost entirely 18-25 year olds who use fitness apps. If available, why not use an online participant recruitment platform like Prolific or Amazon Mechanical Turk? Table 2 is helpful, but I would suggest spending more time explaining the backdrop of the study. The questions/concepts are somewhat clear, but I was unsure what specific scenario or fitness app your participants were thinking about/responding to when answering these questions. Did you ask participants to download a specific app? Or are these questions in reference to an app they’ve already downloaded? Or are you just asking the questions more abstractly?

Response: Dear reviewers, it is necessary to explain to you that the age distribution of the survey samples in the study is in line with the demographic characteristics of the Chinese fitness application users, so the convenience sampling is reasonable. 

Considering the factors such as the survey object and the research area, the research uses the most authoritative online survey platform in China, so the platform of Prolific or Amazon Mechanical Turk is not considered. Hope to get your understanding. Thank you.

Table 2 in the manuscript summarizes the source of the relevant variables and the specific division of the variables. Therefore, in order to better explain the background of its research, we add some original reference variable items in Table 2. Please review it again. Thank you.

In the research sample chapter, the revised manuscript has clarified the research scenario in which the subjects participate, that is, the type of fitness app used. At the same time, in the questionnaire, we did not ask subjects to download specific apps. The setting of the question is based on the subject 's use experience, and the relevant test items are also set up in the questionnaire to identify whether the subject has experience. Only experienced subjects will participate in the survey.

Again, I like the model you outline and report in Figure 3. The question I have here is whether you can explain why flow comes before participation behavior. This point may not be valid if you had explained the study design better, but since I am confused I’ll ask if it is possible for flow experience to happen simultaneously during customer value co-creation. I’m skeptical you can claim mediation here since what you have is not actually an experiment with a manipulated independent variable preceding the dependent variable. I’d suggest an experiment where you experimentally manipulate CKM, then measure flow/attention, and then measure value co-creation.

Response: Dear reviewers, based on literature review and theoretical logic, I have clarified the theoretical connection between flow experience and value co-creation behavior in the second chapter, and explained the research design. Therefore, there is no need to add other experiments in this manuscript. In addition, thank you very much for the research idea of manipulating CKM to verify the relationship between flow experience and value co-creation. At present, I have further carried out relevant research according to your ideas. Thank you very much for your help, thank you.

Improvements in Writing

There are also many instances of typos/inconsistencies/grammatical errors in the manuscript which made it difficult to interpret. The word “APP” is arbitrarily capitalized in some cases but not others. There are hanging spaces in some areas whereas others need a space after a period. A few sentences were not complete. It could also be important to create a stronger hook for the importance of the finding. I’d imagine all fitness app designers strive to delight their customers, leading to enjoyment of their app and thus wanting to co-create. However, what could a designer specifically take away from the survey to specifically enhance their app?

Response: Dear reviewer, Thank you for your comments . I have unified the writing of the word app in the manuscript, and I have also modified the applicability of the hanging space in the manuscript. For some grammatical errors, I also made corresponding corrections. Finally, with regard to the improvement measures for designers ( enterprises ), I have indicated in the 5.1.1 Chapter of the manuscript. Please review it again, thank you.

---

## [Decision Letter · Decision Letter 1]

29 Sep 2024

Relationship between Customer Knowledge Management and the Value Co-Creation of Fitness Application Customers: Mediating Role of Flow Experience

PONE-D-24-07863R1

Dear Dr. wang,

We’re pleased to inform you that your manuscript has been judged scientifically suitable for publication and will be formally accepted for publication once it meets all outstanding technical requirements.

Kind regards,

Hua Pang

Academic Editor

PLOS ONE

---

## [Editor Report · Acceptance letter]

17 Oct 2024

PONE-D-24-07863R1 

PLOS ONE

Dear Dr. Wang, 

I'm pleased to inform you that your manuscript has been deemed suitable for publication in PLOS ONE. Congratulations! Your manuscript is now being handed over to our production team.

Kind regards, 

on behalf of

Dr. Hua Pang 

Academic Editor

PLOS ONE